Biocontrol efficacy of Bacillus licheniformis and Bacillus amyloliquefaciens against rice pathogens

Tariq Mohsin 1 mohsintariq@gcuf.edu.pk
Zahoor Mehvish 1
Yasmeen Tahira 2
Naqqash Tahir 3
http://orcid.org/0000-0002-4941-3469 Rashid Muhammad Abdul Rehman 1 marrashid@gcuf.edu.pk
Abdullah Muhammad 1
Rafiq Abdul Rafay 1
Zafar Marriam 1
Irfan Iqra 1
Rasul Ijaz 1
1 Department of Bioinformatics and Biotechnology, Government College University Faisalabad , Faisalabad , Pakistan
2 Department of Environmental Sciences, Government College University Faisalabad , Faisalabad , Pakistan
3 Institute of Molecular Biology and Biotechnology, Bahauddin Zakariya University , Multan , Pakistan
Mukhtar Prof. Hamid
Electronic publication date: 2025 Jan 29
Publication date: 2025
Volume: 13
Electronic Location ID: e18920
Received 2024 Jul 16; Accepted 2025 Jan 9
Copyright: © 2025 Tariq et al.
Copyright year: 2025
Copyright holder: Tariq et al.
License: This is an open access article distributed under the terms of the Creative Commons Attribution License, which permits unrestricted use, distribution, reproduction and adaptation in any medium and for any purpose provided that it is properly attributed. For attribution, the original author(s), title, publication source (PeerJ) and either DOI or URL of the article must be cited.
License URL: https://creativecommons.org/licenses/by/4.0/

Keywords: Rice, Bacillus, Xanthomonas oryzae, Bipolaris oryzae, 16S rRNA gene

Funding: The authors received no funding for this work.

==============================
Biocontrol is a cost-effective and eco-friendly approach to control plant pathogens using natural enemies. Antagonistic microorganisms or their derivatives specifically target the plant pathogens while minimizing the harm to non-target organisms. Bacterial blight and brown spot are the major rice diseases caused by Xanthomonas oryzae pv. oryzae (Xoo) and Bipolaris oryzae (Bo), respectively. This study was conducted to assess the plant growth-promoting potential and biocontrol activity of root-associated bacteria against the rice pathogens, Xoo and Bo. A total of 98 bacteria were isolated from rice roots and characterized for plant growth-promoting properties including phosphate solubilization, indole-3-acetic acid production, nitrogen fixation and biofilm formation. Based on these properties, 36 bacteria were selected and tested for biocontrol potential against rice pathogens via co-culturing antagonism assay. LE7 exhibited the maximum inhibition of 79%, while FR8, PE2, LE7, LR22 and LR28 also significantly reduced the growth of Xoo. Likewise, FR2, LR22, LR35 and LE7 significantly inhibited the growth of Bo, in which LR22 exhibited the maximum inhibition of 81%. Under controlled-conditions, LE7 and LR22 significantly reduced the disease incidence of Xoo and Bo, respectively, and improved the growth of rice. Full-length 16S rRNA gene sequencing of most potential bacterial isolates, LE7 and LR22, revealed their maximum identity with Bacillus amyloliquefaciens and Bacillus licheniformis, respectively. Application of Bacillus spp. as biocontrol agent represents enormous potential in rice farming. The most promising bacterial isolates could be used as bioinoculants for rice disease management and improved production in a sustainable manner.

Introduction

Phytopathogens are one of the major limiting factors restricting crop productivity. Effective strategies to control phytopathogens can contribute substantially to ensure food security. There are various practices employed to control the damages caused by phytopathogens. In particular, synthetic chemical pesticides provide rapid control of phytopathogens. However, they are non-specific, expensive and contaminating environment (Chatterjee et al., 2023). Conversely, biopesticides and biocontrol agents are host-specific, cost-effective and eco-friendly, offering a promising alternative to conventional synthetic pesticides. Host-specificity of biocontrol agents helps to minimize environmental impact and reduce non-specific effect compared to chemical pesticides (Ayilara et al., 2023). Some microbial biocontrol agents may negatively impact the environment due to broad host range, easier dispersal, incompatibility with the resident microbiota, horizontal gene transfer, etc. (Bonaterra et al., 2012). Therefore, they must be carefully evaluated for risk assessment and non-target effect prior to their release and application in the crop field (De Clercq, Mason & Babendreier, 2011). These biocontrol agents employ a variety of mechanisms such as the production of pest specific toxins, antibiosis, bioantagonism and induction of systemic resistance in host plants (Fira et al., 2018; Vero et al., 2023). Various biocontrol agents exhibit plant growth-promoting activities including phytohormones production, phosphate solubilization, fixing nitrogen to improve soil structure and nutrient-cycling (Meena et al., 2017; Sun et al., 2024). Thus, they develop a symbiotic relationship with plants and increase crop productivity (Pathak et al., 2019). Crops produced by the application of plant growth-promoting and disease controlling microbial agents are recognized as organic crops, which are highly demanded and has great marketability (Akmukhanova et al., 2023; Saldaña-Mendoza et al., 2023).

Oryza sativa L. (rice) is an important crop of great antiquity and akin to progress. It is a rich source of carbohydrates. It provides dietary fiber, various B vitamins (thiamine B1, niacin B3, pyridoxine B6) and minerals such as iron, magnesium and selenium (Caporizzi et al., 2023). Rice production is massively reduced due to various diseases in differnt regions of the world (Rezk et al., 2023). Bacterial blight, blast, brown spot, leaf scald, sheath blight and bakanae disease are the major rice diseases (Conde et al., 2024). Bacterial blight is caused by the Xanthomonas oryzae pv. oryzae (Xoo), which is a major rice disease and reported to cause 20–50% yield loss (Jiang et al., 2020). The symptoms of bacterial blight include drying and yellowing of leaves that starts from the leaf tip and then covers the whole leaf (Fiyaz et al., 2022). Another prominent rice disease is brown spot disease, which is caused by Bipolaris oryzae (Bo) (Marwein et al., 2022). Bo may infect any part of the plant such as leaf, kernel and sheath, at either mature or seedling stage, which results up to 90% yield loss (Chhabra et al., 2023). These are the two major rice diseases in Pakistan (Ul Hassan et al., 2019). Therefore, there is a dire need to develop the biopesticides that can address multiple challenges faced by rice cultivation, offering a cheaper, sustainable and environment-friendly solution. This study was planned to isolate the rice root-associated bacteria, followed by the physiological, biochemical and molecular characterization to evaluate their plant growth-promoting and antagonistic properties.

Materials and Methods

Phytopathogens

The pathogens of bacterial blight, Xoo MXO-3, and brown spot disease, Bo MBO-1, were obtained from Culture Collection Bank, Department of Bioinformatics and Biotechnology, Government College University Faisalabad, Pakistan.

Isolation of bacteria

Six-week old rice plants were collected in sterilized plastic bags from cultivation sites of Faisalabad (31°23′42.5″N and 73°01′45.5″E), Lahore (31°34′55.3620″N and 74°19′45.7536″E) and Pindi-Bhattian (31°53′44.88″N and 73°16′33.77″E), Punjab, Pakistan. Rhizospheric bacteria were isolated as described by Shahid et al. (2014), with some modifications. Plant roots were washed gently to remove the extra soil. Intact roots (1 g) were separated from the plant and placed in a tube containing 9 mL sterilized saline solution (0.85% NaCl). The suspension was mixed well by vortexing and serially diluted upto 10−4. An aliquot of 100 µL from each dilution was spread on LB agar plates and incubated at 28 ± 2 °C for 48 h. Root endophytic bacteria were isolated from the surface sterilized roots. Roots were washed with the sterile water to remove soil, and surface sterilized by immersing in 1% sodium hypochlorite for 2 min and 95% ethanol for 30 s. Roots were rinsed with sterile water three times and crush in autoclaved pastel and mortar containing 1.5 mL saline solution (0.89%). One mL suspension was collected and serially diluted upto 10−4 (Emami et al., 2019). An aliquot of 100 µL from each dilution was spread on LB agar plates and incubated at 28 ± 2 °C for 48 h.

After the incubation, morphologically different bacterial colonies were selected on the bases of color, shape (smooth/irregular), size (small/medium/large), elevation (plain/curved) and appearance (shiny/dull), and purified by sub-culturing (Zahra et al., 2023). Cell morphology and Gram staining of each isolate was performed using light microscope (Wang, Zhu & Chen, 2017). Pure cultures were subjected to the further studies. Pure bacterial cultures were stored in LB broth containing 20% glycerol at −80 °C using 1.5 mL plastic tubes (Rani, Weadge & Jabaji, 2020).

Phosphate solubilization

Phosphate solubilization was tested on Pikovskaya media (Pikovskaya, 1948). A single colony of each bacterial culture was streaked on Pikovskaya plates and incubated at 28 ± 2 °C for 7 days (Azaroual et al., 2020). The plates were observed for halo zone production around the colonies, which is an indicator of P solubilization. Phosphate solubilization activity was calculated as below (Nacoon et al., 2020).

Phosphatesolubilizationindex(PSI)=halozonediameter/colonydiameter

IAA production

IAA production was quantified by calorimetric method using Salkowski’s reagent. Bacterial isolates were grown in LB broth supplemented with tryptophan (100 µg L−1) and incubated at 28 ± 2 °C for 72 h. Bacterial cultures were centrifuged for 10 min at 12,000 rpm. An aliquot of 1 mL of supernatant was vigorously mixed with 4 mL of Salkowski’s reagent (5 mL 0.5M FeCl3. 6H2O, 150 mL conc. H2SO4, 250 mL H2O) and incubated for 30 min (Gordon & Weber, 1951). Absorbance was recorded at 530 nm using a spectrophotometer (MRC Laboratory Instruments UV-1200; MRC Laboratory Instruments, Cambridge, UK). Quantity of IAA produced by the bacterial isolates was calculated by comparing data with a standard curve generated from the IAA standard solutions (Zhang et al., 2021).

Nitrogen fixation

Nitrogen fixation activity of the potential bacterial isolates was determined by growing bacteria on nitrogen-free mineral (NFM) media (Wang et al., 2020). Bacterial isolates were streaked on the NFM agar plates and incubated at 28 °C for 48 h. After the incubation, the growth of bacteria was observed as a representation of nitrogen fixation ability, which was measured as arbitrary values i.e., strong (+++), moderate (++), weak (+) and no (−) (Hardarson, Golbs & Danso, 1989; Mirza & Rodrigues, 2012).

Biofilm formation

Biofilm formation was studied according to Zahra et al. (2023) with some modifications. Bacterial cultures were grown in LB broth, pelleted by centrifugation at 8,000 rpm for 2 min and washed with sterile distilled water. The cells were resuspended in the same medium and diluted to 0.2 OD600 nm. An aliquot of 200 µL of each bacterial isolates was added to each well in a 96-well polyvinyl chloride (PVC) plate with six replicates. LB alone was used as a control. The plates were covered with plastic lids and incubated at 28 °C for 24 h. Bacterial culture was removed from the wells after the incubation period and the plates were washed gently with sterile water. The plates were allowed to dry, and the wells were treated with 200 µL of 0.001% crystal violet for 15 min. The excess dye was removed and the wells were washed with sterile water. An aliquot of 200 µL of 95% ethanol was added into each well to solubilize the stain adhered to the wall. Biofilm formation ability was recorded by quantifying the blue coloration at an absorbance 570 nm in a plate reader.

Antagonism assay

Potential of bacterial isolates to control Xoo was tested by overlay plate technique according to Mrabet et al. (2006), with some modifications. Log phase grown bacterial isolates were diluted to 104 cfu mL−1 in sterile water and 3 mL of suspension was mixed in molten hand cool 25 mL LB agar and poured in petri plate. Bacterial isolate was replaced with sterile H2O in negative control treatments. After solidification, 5 μL of concentrated culture (109 cfu mL−1) of Xoo was inoculated in the center of each plate. Plates were incubated at 28 ± 2 °C for 72 h and radii of pathogen growth were measured in all the treatments.

Potential of bacterial isolates to control Bo was tested by dual culture technique according to Tariq, Yasmin & Hafeez (2010), with some modification. A disk of 5 mm2 of pure Bo culture was placed at the center of a Petri dish containing potato dextrose agar (PDA). Pathogen plates were prepared for all selected 36 isolates and a control. A circular line of 4 cm diameter of each bacterial suspension (2 × 109 cfu mL−1) was streaked surrounding the pathogen disk. Bacterial suspension was replaced with sterile H2O in negative control treatments. Plates were incubated at 28 ± 2 °C for 72 h and radii of pathogen growth were measured in all the treatments.

The results of antagonism assay were expressed as % inhibition of pathogen growth using the following formula (Tariq, Yasmin & Hafeez, 2010; Gonzalez et al., 2020).

%inhibition=[1−(pathogengrowthinbacterialtreatment/pathogengrowthincontroltreatment)]×100.

Controlled-conditions experiment

A controlled-conditions experiment was conducted to evaluate the biocontrol potential of five selected bacterial isolates (LR22, LR28, PE2, LE7 and FR8) against Xoo. Culture of Xoo was grown in 100 mL LB broth at 28 °C for 48 h. Culture was centrifuged at 6,000 rpm for 5 min, supernatant was discarded, pellet was dissolved in 100 mL sterilized water, and this solution was mixed in 12 kg autoclaved soil. Each plastic pot was filled with 500 g of pathogen-infested soil. Pot experiment was conducted in a completely randomized design (CRD) with four replicates and six treatments (LR22, LR28, PE2, LE7, FR8 and negative control containing sterile distilled water). Rice seeds (Cultivar 1121) were surface sterilized with 2% sodium hypochlorite for 1 min and washed thrice with sterilized water (Kaga et al., 2009). Surface sterilized seeds were immersed for 5 min in bacterial treatments diluted to OD 600 nm = 0.5. Seeds were sown in pathogen-infested soil and watered with sterile distilled water Pots were placed in a growth chamber adjusted at 30 ± 2 °C for 16 h light period and 20 ± 2 °C for 8 h dark period. After 6 weeks, the plants were harvested and various parameters such as disease incidence, biocontrol efficiency, plant fresh weight and plant dry weight, were recorded. Disease incidence was recorded by counting the number and size of disease spots on the stem and leaves. Biocontrol efficacy was recorded using the following formula (Tariq, Yasmin & Hafeez, 2010; Yasmin et al., 2017).

Biocontrolefficacy=[(diseaseincidenceofcontrol–diseaseincidenceoftreatment)/diseaseincidenceofcontrol]×100.

Another controlled-conditions experiment was conducted to evaluate the biocontrol potential of four selected bacterial isolates (FR2, LR22, LR35 and LE7) against Bo, with four replicates. Culture of Bo was grown in 100 mL potato dextrose broth (PDB) at 28 °C for 96 h. Culture was centrifuged at 6,000 rpm for 5 min, supernatant was discarded, pellet was dissolved in 100 mL sterilized water, and this solution was mixed in 10 kg autoclaved soil. All the other procedure was same as mentioned in the above experiment (Tariq, Yasmin & Hafeez, 2010).

All the data sets were analyzed for statistical variance and means comparison using least significant difference test at P = 0.05. The data was analyzed statistically using CoStat software version 6.303 (Cardinali & Nason, 2013).

16S rRNA gene amplification and sequencing

Most promising antagonistic bacteria were further identified based on the sequencing of 16S rRNA gene according to the La Pierre et al. (2017), with some modifications. 16S rRNA gene was amplified using universal primers fD1 (5-AGAGTTTGATCCTGGCTCAG-3) and rD1 (5-AAGGAGGTGATCCAGCC-3), which amplified 1,500 bp fragment (Weisburg et al., 1991). Amplified PCR products of 16S rRNA gene were separated on 1% agarose. Amplified PCR products of 16S rRNA gene were purified using FavorPrep PCR Clean-Up Mini Kit (FAVORGEN) according to the standard protocol recommended by the manufacturer. PCR products were sequenced using the services of Macrogen Inc. Korea. DNA sequences were compared with other sequences available in databases connected with NCBI BLAST tool (Altschul et al., 1990). Sequences were submitted to the NCBI GenBank and accession numbers were obtained. A phylogenetic tree was constructed by Mega 11 software using the 16S rRNA gene sequences of this study and highly related type strains sequences retrieved from DNA database with 1,000 bootstrap value (Kumar, Stecher & Tamura, 2016). Pairwise identity chart was constructed using the Sequence Demarcation Tool (SDT) to provide an insight into the similarities and differences among the sequences of selected species (Kumar, Stecher & Tamura, 2016; Tariq et al., 2023).

Results

Isolation of bacteria

A total of ninety-eight (98) bacterial morphotypes were isolated from rhizosphere and root interior of rice samples. Twenty-eight bacteria were isolated from rice samples of Faisalabad, out of which 17 were rhizospheric and 11 were endophytic. Fifty-three bacteria were isolated from rice samples of Lahore, out of which 36 were rhizospheric and 17 were endophytic. Seventeen bacteria were isolated from rice samples of Pindi-Bhattian, out of which 10 were rhizospheric and seven were endophytic. Collectively, 63 were rhizospheric and 35 were endophytic bacteria. Colony and cell morphological characters were highly variable among the bacterial isolates. Colony morphology, cell morphology and Gram staining of all the bacterial isolates are mentioned in Table S1.

Plant growth-promoting attributes

All bacterial isolates were screened for phosphate solubilization on Pikovskaya’s agar medium. Thirty-three bacterial isolates were found to exhibit phosphate solubilization ability. FR5, LR22 and PR3 displayed better phosphate solubilization ability by producing index of 4.03, 4.0 and 4.8, respectively (Table 1). In case of IAA production, only 29 bacterial isolates exhibited IAA production. FE11, LE4 and PE7 demonstrated better potential of IAA production with 13.46, 14.1 and 13.5 µg IAA mL−1, respectively (Table 1). Bacterial isolates, FE11, LE4, LE7, PR3 and PE7, showed significant nitrogen fixation (Table 1). Additionally, twenty-five bacterial isolates showed biofilm formation ranging from 0.25–3.2 at OD570 nm. PE7 exhibited the highest biofilm formation compared to all other isolates (Table 1).

Table 1 Quantification of plant growth-promoting properties exhibited by the potential bacterial isolates.

Site	Isolate	Phosphate solubilization (Index)	IAA production (µg mL−1)	Nitrogen fixation	Biofilm formation (OD 570 nm)	
Faisalabad	FR1	2.1 ± 0.32	0		0.3 ± 0.12	
FR2	0	6.6 ± 0.29	+	1.9 ± 0.55	
FR3	2.0± 0.18	4.4 ± 0.5		1.0 ± 0.16	
FR5	4.0 ± 0.17	0 ± 0		0.8 ± 0.10	
FR6	1.0 ± 0.25	5.4 ± 0.37	++	0.7 ± 0.14	
FR8	2.7 ± 0.10	4.4 ± 0.37		0.6 ± 0.03	
FR10	1.5 ± 0.43	2.5 ± 0.43		0	
FR11	0	5.8 ± 0.22	++	0.3 ± 0.1.0	
FR17	2.0± 0.30	0		0.5 ± 0.02	
FE6	2.1 ± 0.33	4.0 ± 0.35	++	1.2 ± 0.08	
FE10	1.6 ± 0.25	0		1.6 ± 0.11	
FE11	2.0 ± 0.28	13.4 ± 0.36	+++	0	
Lahore	LR5	2.0 ± 0.35	7.3 ± 0.32	++	0	
LR8	2.2 ± 0.36	3.4 ± 0.31		0	
LR13	2.0 ± 0.20	4.0 ± 0.5		0	
LR15	2.8 ± 0.21	6.8 ± 0.6	++	0.83 ± 0.17	
LR17	0.9 ± 0.21	2.4 ± 0.37		1.6 ± 0.59	
LR21	2.1 ± 0.53	2.7 ± 0.53		0	
LR22	4.0 ± 0.49	8.7 ± 0.44	+	2.1 ± 0.32	
LR23	2.8 ± 0.5	8.5 ± 0.40	++	0.6 ± 0.10	
LR26	2.9 ± 0.1	0		0.9 ± 0.14	
LR28	2.5 ± 0.5	4.1 ± 0.36		0	
LR31	1.0 ± 0.22	6.1 ± 0.56		0	
LR35	3.3 ± 0.35	2.0 ± 0.33		0.5 ± 0.07	
LE4	1.9 ± 0.45	14.1 ± 0.42	+++	0.69 ± 0.07	
LE5	2.1 ± 0.0.5	9.9 ± 0.64		0	
LE7	3.5 ± 0.42	12.9 ± 0.26	+++	3.1 ± 0.4	
LE17	0.7 ± 0.14	1.1 ± 0.08		0.5 ± 0.07	
Pindi Bhattian	PR2	4.8 ± 0.28	0		1.7 ± 0.48	
PR3	4.8 ± 0.21	2.3 ± 0.30	+++	0	
PR7	1.8 ± 0.24	0		0.5 ± 0.05	
PR8	2.4 ± 0.34	3.9 ± 0.22	+	0.3 ± 0.19	
PR10	2.2 ± 0.35	2.8 ± 0.25		1.3 ± 0.17	
PE2	1.8 ± 0.14	3.6 ± 0.36		0	
PE3	3.9 ± 0.47	2.2 ± 0.10	++	0.2 ± 0.09	
PE7	2.4 ± 0.2	13.5 ± 0.41	+++	3.2 ± 0.30	

Antagonism assay

Thirty-six potential bacterial isolates were selected and checked for antagonism against Xoo and Bo in in vitro plate assay. Twenty-eight isolates showed antagonistic activity against Bo and thirty-three showed antagonism against Xoo at varying levels (Table 2). Bacterial isolate LE7 exhibited maximum antagonistic activity by inhibiting 79% growth of Xoo (Fig. 1). LR22 exhibited maximum antagonistic activity by inhibiting 81% growth of Bo (Fig. 2). Growth of pathogens was clearly observed in negative control.

Table 2 Growth inhibition of Xanthomonas oryzae pv. oryzae and Bipolaris oryzae by rice root-associated bacteria.

Bacterial isolates	Xoo growth inhibition (%)	Bo growth inhibition (%)	
FR1	58.1 ± 4.82	53.1 ± 4.07	
FR2	59.4 ± 5.73	70.8 ± 2.67	
FR3	0	0	
FR5	58.6 ± 6.38	44.8 ± 2.29	
FR6	42.3 ± 2.54	45.5 ± 4.01	
FR8	68.8 ± 4.5	1.6 ± 0.56	
FR10	17.6 ± 2.37	5.8 ± 1.12	
FR11	42.1 ± 3.41	60.8 ± 3.10	
FR17	43.5 ± 3.22	0	
FE6	34.0 ± 3.56	0	
FE10	0	0	
FE11	43.2 ± 3.64	61.4 ± 3.28	
LR5	46 ± 2.05	41.1 ± 2.34	
LR8	32.5 ± 3.85	59.4 ± 2.92	
LR13	52.9 ± 2.10	64.6 ± 2.43	
LR15	39.9 ± 2.79	60.3 ± 3.53	
LR17	48.1 ± 2.61	0	
LR21	20.4 ± 2.40	52.2 ± 4.57	
LR22	68.9 ± 3.57	81.5 ± 4.34	
LR23	33.4 ± 4.40	20.8 ± 1.80	
LR26	51.7 ± 3.3	44.5 ± 2.16	
LR28	61.5 ± 5.75	0	
LR31	41.6 ± 3.76	0	
LR35	39.6 ± 3.33	73.5 ± 4.72	
LE4	43.3 ± 3.96	32.5 ± 3.34	
LE5	32.7 ± 4.53	13 ± 2.61	
LE7	78.8 ± 3.33	71.1 ± 2.14	
LE17	51.6 ± 6.93	63.3 ± 3.27	
PR2	47.8 ± 3.87	32.4 ± 2.78	
PR3	52.5 ± 3.55	49.1 ± 2.95	
PR7	17.3 ± 2.41	2.3 ± 0.77	
PR8	12.2 ± 1.94	48.2 ± 3.93	
PR10	53.3 ± 3.57	40.8 ± 3.92	
PE2	69.3 ± 3.046	46.4 ± 4.13	
PE3	21.3 ± 2.33	63.8 ± 2.58	
PE7	14.7 ± 4.17	0	

Figure 1 Biocontrol of Xanthomonas oryzae pv. Oryzae (Xoo).

Bacterial isolate LE7 significantly inhibited the growth of Xoo (left side Petri plate). Xoo grew well in the absence of bacteria (right side Petri plate).

Figure 2 Biocontrol of Bipolaris oryzae (Bo).

Bacterial isolate LR22 significantly inhibited the growth of Bo (left side Petri plate). Bo grew well in the absence of bacteria (right side Petri plate).

Controlled-conditions experiment

In pot assay of Xoo infested soil, FR8 exhibited the lowest biocontrol efficiency (16.3%), whereas LE7 demonstrated the highest biocontrol efficiency (89.4%) against Xoo. Moreover, an increase in plant fresh weight was observed across the treatments, ranging from 228.8 mg for FR8 to 308.3 mg for LE7. Similarly, the maximum increase of 60.6% in plant dry weight was observed upon the application of LE7, compared to control (Table 3).

Table 3 Effect of antagonistic bacterial isolates on Xanthomonas oryzae pv. oryzae treated rice plants.

Bacterial isolates	Disease incidence (%)	Biocontrol efficiency (%)	Plant fresh weight (mg)	Plant dry weight (mg)	
LR22	31.7 ± 3.24 b	18.5 ± 8.3 c	239 ± 20.04 cd	37.8 ± 3.41 bc	
LR28	10.7 ± 2.23 cd	71.9 ± 5.48 ab	289.5 ± 9.12 ab	48.3 ± 1.9 ab	
PE2	17.7 ± 1.66 c	54.3 ± 4.56 b	254.2 ± 23.2 bc	43.4 ± 6.24 ab	
LE7	4 ± 1.24 d	89.3 ± 3.59 a	308.2 ± 19.79 a	49.9 ± 0.58 a	
FR8	32 ± 3.05 b	16.3 ± 3.57 c	228.7 ± 18 cd	39.9 ± 4.24 abc	
Control	39.5 ± 3.93 a	–	192.5 ± 19.28 d	31.1 ± 2.52 c	
LSD	7.01	23.03	46.21	10.78	
ANOVA significance	***	***	**	*	
Notes:

LSD, least significant difference.

Each value represents mean (n = 4) ± standard error.

Values followed by the different letters in same column indicate significant difference.

* Represents significant.

** Represents moderately significant.

*** Represents highly significant.

In pot assay of Bo infested soil, highly significant influence of bacterial treatments was observed on selected parameters of rice growth. The biocontrol efficiencies against Bo varied across the treatments, ranging from 40.3% for FR2 to 84.6% for LR22. Additionally, there was an increase in plant fresh weight, ranging from 220.8 mg for FR2 to 330.8 mg for LR22. Likewise, the maximum increase of 74.7% in plant dry weight was recorded in LR22, compared to control (Table 4).

Table 4 Effect of antagonistic bacterial isolates on Bipolaris oryzae treated rice plants.

Bacterial isolates	Disease incidence (%)	Biocontrol efficiency (%)	Plant fresh weight (mg)	Plant dry weight (mg)	
FR2	26.5 ± 2.6 b	40.3 ± 6.82 c	220.7 ± 10.14 cd	36.3 ± 2.44 bc	
LR22	7 ± 0.81 d	84.5 ± 0.73 a	330.7 ± 17.01 a	50.7 ±1.94 a	
LR35	23 ± 2.05 bc	47.5 ± 7.32 bc	238 ± 16.63 c	37.2 ± 3.59 b	
LE7	17.5 ± 1.97 c	60.7 ± 4.79 b	275.5 ± 16 b	43.5 ± 2.56 ab	
Control	45.2 ± 4.25 a	–	185.5 ± 7.43 d	29 ± 1.25 c	
LSD	6.77	17.14	36.52	7.49	
ANOVA significance	***	***	***	***	
Notes:

LSD, least significant difference.

Each value represents mean (n = 4) ± standard error.

Values followed by the different letters in same column indicate significant difference.

*** Represents highly significant.

Identification of potential bacteria

Most efficient bacterial isolates, LR22 and LE7, exhibiting biocontrol and plant growth-promoting activities were taxonomically identified by sequencing 16S rRNA gene. LR22 and LE7 showed high similarity with multiple Bacillus species. However, LE7 showed maximum similarity with Bacillus amyloliquefaciens and LR22 showed maximum similarity with Bacillus licheniformis. Their sequences were submitted to the Genbank NCBI under the accession number OP143931 and OP143932, respectively. A phylogenetic tree of the potential isolates with type strains further strengthened that LE7 is B. amyloliquefaciens and LR22 is B. licheniformis. The selected sequences were grouped into multiple clades in the phylogenetic tree (Fig. 3). A color coded pairwise identity matrix was constructed, in which each colored cell represents the percentage identity of two sequences. The selected sequences showed an identity ranging from 77–100% (Fig. 4).

Figure 3 Taxonomic positioning of bacterial species using 16S rRNA gene sequences.

Phylogenetic tree based on 16S rRNA gene sequence of rice root-associated antagonistic bacteria identified in current study are yellow highlighted. The tree was arbitrarily rooted on the sequence of the Aquifex pyrophilus Kol5a (NR029172.1) with red highlight. Values at nodes indicate percentage bootstrap values.

Figure 4 Pairwise identity chart of bacterial species using 16S rRNA gene sequences.

Sequences of selected bacteria showed identity percentages ranged 77–100.

Discussion

The aim of this study was to isolate and characterize the rice endophytic and rhizospheric bacteria for plant growth promotion and antagonistic potential against rice pathogens, Xoo and Bo. In total, 98 root-associated bacteria were isolated. Out of which, 63 were rhizospheric and 35 were endophytic. All the bacterial isolates were initially tested for plant growth-promoting potential such as phosphate solubilization, nitrogen fixation, IAA production, biofilm formation and biocontrol ability. Further, in vivo testing revealed that two bacterial isolates, LR22 and LE7, showed maximum plant growth-promoting and biocontrol activity against rice pathogens. 16S rRNA gene sequencing revealed maximum similarity of LE7 with Bacillus amyloliquefaciens and LR22 with Bacillus licheniformis. B. licheniformis has previously been isolated from rhizospheric soil and roots of different plant species including rice (Damodaran et al., 2018; Liu et al., 2021; Rawat et al., 2022; Zhan et al., 2023), Coffea arabica (James et al., 2023), Vigna radiata (Bhutani et al., 2022), camel feces (Qi et al., 2023), Collema undulatum lichens (Medison et al., 2023) and the heat-stressed spinach (Li et al., 2023). Likewise, B. amyloliquefaciens has previously been isolated from the wild rice root endosphere (Tian et al., 2023), rhizosphere soil of healthy apple trees (Duan et al., 2021) and wheat (Xu et al., 2021).

P is an essential macronutrient vital for plant growth, cell division, photosynthesis and energy production (Khan et al., 2010). Both B. licheniformis LR22 and B. amyloliquefaciens LE7 belonging to current study were proved to be efficient P solubilizers. Previously, Won et al. (2019) reported an increase in total P contents of soil by the application of B. licheniformis MH48. Ahmed et al. (2022) also observed P solubilizing activity of B. amyloliquefaciens on Pikovskaya medium. Mao et al. (2022) demonstrated the P solubilizing potential of B. licheniformis, which can potentially be employed to prevent and control phosphorus-based diseases in potato plants. In the current investigation, B. licheniformis LR22 and B. amyloliquefaciens LE7 exhibited better IAA production. IAA is the most important plant hormone that plays a significant role in diverse aspects of plant physiology and development. It is involved in cell growth, root development, tropisms, fruit development, flowering, senescence, abscission and stress responses (Fu et al., 2015). A significant proportion of soil bacteria possess the ability to produce IAA (Mohite, 2013). IAA produced by plant growth-promoting bacteria (PGPB), plays a vital role in root hair formation and aids plants in withstanding drought stress (Ali & Khan, 2021). Panichikkal et al. (2021) also highlighted IAA producing ability of B. licheniformis in a study involving Capsicum annuum (chili) seedlings. Mahdi et al. (2020) determined the ability of B. licheniformis to produce IAA in TSB medium supplemented with 0.1% L-tryptophan as a metabolic precursor for IAA synthesis. Shao et al. (2021) discussed the involvement of the ysnE gene in IAA biosynthesis in B. amyloliquefaciens SQR9. Putrie, Aryantha & Antonius (2021) reported the presence of IAA n-acetyltransferase, an enzyme involved in IAA synthesis, in B. amyloliquefaciens. According to Ji et al. (2021), B. amyloliquefaciens strain Ba13 has the ability to synthesize IAA through the IpyA pathway. The presence of genes including patB, yclC (phenolic acid decarboxylase) and dhaS in the genome of strain Ba13, indicated the potential for IAA biosynthesis via IpyA pathway. Nitrogen fixation is the conversation of atmospheric nitrogen into the form that can be utilized by plants (Javed et al., 2022; Raghuram et al., 2022). Nitrogen is a vital macronutrient for plants supporting protein synthesis, chlorophyll production, enzyme activity, growth, nutrient assimilation, stress tolerance and reproductive processes. Its availability and proper utilization are essential for optimal plant performance and sustainable agriculture (Javed et al., 2022; Raghuram et al., 2022). In this study, both B. licheniformis LR22 and B. amyloliquefaciens LE7 showed N fixation potential. Our findings are in agreement with Yousuf et al. (2017), who isolated nitrogen fixing B. licheniformis from a tropical estuary and adjacent coastal sea and attributed its nitrogen fixing potential to the nifH gene. According to Won et al. (2019) B. licheniformis MH48 significantly increased the total nitrogen content of the soils compared to the control group, indicating its ability to fix atmospheric nitrogen. Kazerooni et al. (2021) also confirmed that B. amyloliquefaciens isolated from Sasamorpha borealis exhibited ability to fix nitrogen on Nitrogen-free bromothymol blue malate (NFb) medium. Cui et al. (2022) identified endophytic B. amyloliquefaciens exhibiting nitrogen fixation ability and potential to control potato scab disease.

B. licheniformis is a potent biocontrol agent, producing a variety of antimicrobial compounds to combat plant pathogens and pests. A study conducted by Nigris et al. (2018) indicated that B. licheniformis has the potential to suppress grapevine (Vitis vinifera cv. Glera) fungal pathogens and inhibits their growth through diffusible molecules. Liu et al. (2021) also demonstrated the biocontrol effect of B. licheniformis against Magnaporthe oryzae by reducing chitin contents and maintaining integrity of the cell wall. According to Ul Hassan et al. (2019), B. licheniformis effectively controls Aspergillus flavus, prevents its sporulation and inhibits aflatoxin accumulation on maize ears and acts as a biocontrol agent against mycotoxigenic fungi in cereal grain storage.

B. amyloliquefaciens is also a promising biological control agent against agricultural pests and pathogens. It produces antifungal compounds, exhibits bactericidal properties and inhibits a wide range of pathogens (Salazar, Ortiz & Sansinenea, 2017). Lee et al. (2015) found that B. amyloliquefaciens, isolated from wild ginseng, demonstrated effective biocontrol activity against Phytophthora cactorum. Wu et al. (2015) also reported that biocontrol activity of B. amyloliquefaciens against Xoo and X. oryzae pv. oryzicola, the causative agents of bacterial blight and bacterial leaf streak diseases in rice. This biocontrol activity was attributed to the production of two antibiotic compounds i.e., difficidin and bacilysin.

The ability to form biofilms is a survival strategy employed by many microorganisms inhabiting harsh environments. A biofilm is a complex community of microorganisms that adhere to surfaces and are embedded in a self-produced matrix of extracellular polymeric substances (Muhammad et al., 2020). In this study, all the rice root-associated bacteria demonstrated biofilm forming potential at varying levels. In particular, B. licheniformis LR22 and B. amyloliquefaciens LE7 exhibited an exceptional ability to form biofilms, as evident by their robust optical density measurements. Our results are in agreement with Yasmeen et al. (2020), who reported the occupancy of biofilm forming B. licheniformis in root and suggested its use for plant growth promotion and stress alleviation in sunflower under salinity stress conditions. Similarly, Wang et al. (2019) suggested that biofilm formation of B. amyloliquefaciens plays a crucial role in root colonization and induced drought tolerance in tomato plants.

Biopesticidal and plant growth-promoting potential of B. licheniformis and B. amyloliquefaciens were further assessed by pot experiment under controlled-conditions. Both of these bacterial isolates exhibited strong biocontrol activity against Bo and Xoo, reduced disease incidence and improved plant growth. These isolates could employ diverse mechanisms to control phytopathogens while sustaining environmental health. Microbial biopesticides can produce pest-specific toxins that target pests’ gut receptors, causing paralysis and death without harming non-target organisms. Another mechanism called antibiosis involves antimicrobial compounds that disrupt pathogen’s physiology, reducing its survival and reproduction (Aslam et al., 2024; Aziz et al., 2024). Through antagonism, beneficial microbes outcompete pathogenic ones, lowering disease incidence. Biopesticides also induce systemic resistance in plants, activating defense pathways to robust resistance. Compared to synthetic pesticides, biopesticides offer environmental safety, target specificity, and support sustainable agricultural practices by reducing chemical reliance and promoting biodiversity (Fira et al., 2018; Vero et al., 2023).

Rawat et al. (2022) reported that sole application or consortium of B. licheniformis, Pantoea dispersa and Staphylococcus sp. significantly improved rice seed germination percentage, seedling vigor index and various root traits including root length, root volume, root dry weight, root projected area, root tip number, root segment number, root volume and average root diameter. Ghazy & El-Nahrawy (2020) reported that B. licheniformis exhibited biocontrol activity against the maize-infecting Cephalosporium maydis. Moreover, the findings of Abo-Elyousr et al. (2022) indicated that B. licheniformis, along with B. aerius, showed effectiveness against cucumber powdery mildew under both in vitro and in vivo conditions. B. licheniformis demonstrates biocontrol abilities by significantly reducing Meloidogyne incognita (root-knot nematode) infection in tomato plants. Additionally, it promotes plant growth through the production of volatile substances that induce stomatal closure, enhance plant system tolerance and regulate gene expression related to cell wall structure. The application of B. licheniformis also influences the composition of microbial communities in the rhizosphere, contributing to plant health and productivity (Du et al., 2022).

Ji et al. (2021), also observed significant improvements in different growth parameters of tomato such as stem branching number, stem width and plant fresh weight upon the inoculation of B. amyloliquefaciens compared to the control group across different growth stages. Chu et al. (2021) identified B. amyloliquefaciens strain as a potential biocontrol agent for managing bacterial wilt caused by Ralstonia solanacearum and its positive role in plant growth promotion. Cui et al. (2019) indicated that B. amyloliquefaciens isolated from wheat rhizosphere, showed promising potential as a biocontrol and plant growth-promoting agent against southern corn leaf blight caused by Bipolaris maydis. A study conducted by Yuan et al. (2014) also revealed that B. amyloliquefaciens significantly suppressed the incidence of Fusarium wilt disease as well as promoted the growth of banana plants by producing phytohormones like GA3 and IAA and phytase enzyme.

Conclusion

This study was aimed to isolate and characterize endophytic and rhizospheric bacteria for their potential to improve rice growth and antagonizing its pathogens, Xoo and Bo. Out of 98 bacterial isolates, Bacillus amyloliquefaciens LE7 and Bacillus licheniformis LR22 demonstrated tremendous in vitro potential to exhibit biocontrol activity against rice pathogens besides their ability to solubilize phosphate, fix nitrogen, produce IAA and form biofilm. These traits are important in promoting plant growth and protecting plants from diseases. Under controlled-conditions, B. amyloliquefaciens LE7 and B. licheniformis LR22 significantly reduced the disease incidence and improved plant growth. These bacterial isolates could be used as biopesticides and biofertilizers after field-testing for improved production of rice.

Supplemental Information

Supplemental Information 1 Colony morphology, cell morphology and Gram staining of all the rice root-associated bacteria.

Supplemental Information 2 PGPR characteristics raw data.

Supplemental Information 3 Inhibition of pathogens raw data.

Supplemental Information 4 Antagonism pot assay raw data.

Additional Information and Declarations

Competing Interests

Mohsin Tariq is an Academic Editor for PeerJ.

Author Contributions

Mohsin Tariq conceived and designed the experiments, performed the experiments, analyzed the data, prepared figures and/or tables, authored or reviewed drafts of the article, and approved the final draft.

Mehvish Zahoor conceived and designed the experiments, performed the experiments, analyzed the data, authored or reviewed drafts of the article, and approved the final draft.

Tahira Yasmeen performed the experiments, prepared figures and/or tables, and approved the final draft.

Tahir Naqqash performed the experiments, prepared figures and/or tables, and approved the final draft.

Muhammad Abdul Rehman Rashid conceived and designed the experiments, prepared figures and/or tables, and approved the final draft.

Muhammad Abdullah conceived and designed the experiments, prepared figures and/or tables, authored or reviewed drafts of the article, and approved the final draft.

Abdul Rafay Rafiq conceived and designed the experiments, authored or reviewed drafts of the article, and approved the final draft.

Marriam Zafar conceived and designed the experiments, analyzed the data, authored or reviewed drafts of the article, and approved the final draft.

Iqra Irfan performed the experiments, analyzed the data, authored or reviewed drafts of the article, and approved the final draft.

Ijaz Rasul analyzed the data, authored or reviewed drafts of the article, and approved the final draft.

DNA Deposition

The following information was supplied regarding the deposition of DNA sequences:

Data are available at GenBank: OP143931–OP143932.

Data Availability

The following information was supplied regarding data availability:

Raw data, including colony morphology, cell morphology and Gram staining of all the rice root-associated bacteria, are available in the Supplemental Files.

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
