# Peer review of "Biocontrol efficacy of Bacillus licheniformis and Bacillus amyloliquefaciens against rice pathogens"

_PeerJ, doi:10.7717/peerj.18920_

## Round 0.1 · original submission · Major Revisions

Please address all the queries raised by the reviewers.

Special attention be given to the English grammar and spellings.

I recommend change in title to make it more precise and inclusive.

·

Basic reporting

The manuscript presents valuable research on biocontrol agents and their application in rice cultivation, which is highly relevant and significant. The study is methodologically sound and provides detailed data, which supports its findings. However, there are some drawbacks which need to be fixed.
The title needs revision to make it concise and robust.
There are many economically rice pathogens but the authors dealt with only two. This should be justified.
Streamline the abstract by focusing on the main findings and their significance. Some terms should be more precise, such as “superlative isolates,” which is vague.
Revise the introduction by removing redundant information and focusing on the key points. Streamline references to the most pertinent studies to maintain focus. The authors are suggested to read and potentially cite the following latest articles to strengthen the introduction:
Aslam, N., Atiq, M., Rajput, N.A., Akram, A., Arif, A.M., Kachelo, G.A., Nawaz, A., Jahangir, M.M., Jabbar, A. and Ijaz, A., 2024. Explicating Botanical Bactericides as an Intervention Tool towards Citrus Canker. Plant Protection, 8(1), 25-32.
Aziz, S., Jamshed, S.A., Mukhtar, T., Irshad, G., Ijaz, S.S. and Raja, M.U. 2024. Evaluation of Bacillus spp. as biocontrol agents against chili leaf spot caused by Xanthomonas vesicatoria. Journal of Plant Diseases and Protection, 131: 987–997 https://doi.org/10.1007/s41348-024-00866-5

Experimental design

Some sections in the materials and methods, such as the descriptions of media compositions, are overly detailed. The methodology should be structured more clearly.

Validity of the findings

The results section is dense and difficult to follow due to the extensive detail. Some data presentation lacks clarity, such as in the description of biocontrol efficiency.
The discussion could be better structured by grouping related findings together. For example, all information related to plant growth promotion could be presented in one section, followed by sections on biocontrol activity, biofilm formation, and so forth. This would improve readability and clarity.
The authors should describe in detail the inferences of their findings. For example, what are the potential mechanisms by which the Bacillus strains promote plant growth and suppress disease? How do the findings contribute to the understanding of plant-microbe interactions?
The authors could discuss the limitations of the study and suggest future research directions to address these limitations.
The conclusion should be strengthened by highlighting the potential practical applications of the findings. For example, the authors should discuss the potential use of the Bacillus strains as biofertilizers or biopesticides. The authors should also briefly mention the broader effects of the research for sustainable agriculture and food security.

Additional comments

The language is generally good, but some sentences are complex and could be broken down for clarity.

Reviewer 2 ·

Basic reporting

The manuscript describes the identification and characterization of 98 bacteria isolated from the roots of rice plants grown in soil for the control of bacterial blight and brown spot diseases caused by Xanthomonas oryzae pv. oryzae (Xoo) and Bipolaris oryzae (Bo), respectively. Screening of these isolates revealed that seven showed inhibitions of Xoo and/or Bo growth in vitro. Two of them, LR22 and LE7, were found to be capable of improving rice growth and reducing the disease incidence.

Experimental design

Overall, the manuscript is clearly written, and provides some useful information of the bacterial isolates that could be used for disease control in a major food crop.

A major concern for this manuscript is the lack of sufficient information on some of their studies. For example, how many plants were used in the “controlled conditions experiment” to prove that LR22 and LE7 can reduce the disease incidence? The manuscript would benefit significantly from more detailing of the experiments. Figures showing diseased as well as healthy plants would also be helpful.

Abbreviations (e.g., Xoo) should be used across the manuscript.

Validity of the findings

The authors should confirm their key observations by more rigorous experiments. For example, manually inoculate (e.g., by the leaf clipping method) the plants treated with LR22 and LE7 to show resistance to Xoo can be established.

---

## Round 0.2 · Minor Revisions

Please address all the queries raised by the reviewers especially reviewer 3 and 4. Also please have the whole manuscript read by an expert in English to remove all the language issues present in the MS.

·

Basic reporting

Modified

Experimental design

Changes made

Validity of the findings

Compliance made

Additional comments

The suggested changes were addressed positively, including revisions to the title, abstract, and introduction. Methodology details were streamlined, the results section was clarified, and the discussion was restructured. Limitations, future research directions, and practical applications were included, enhancing the overall manuscript.

Reviewer 3 ·

Basic reporting

The manuscript is a resubmission based on previous reviewers comments. In general, the results explore the potential of biocontrol of a series of isolated microorganisms from the rhizosphere of 6-week rice plants. Literatura is mostly adequate where few positive agronomic uses could be more broadly presented. On another note, a few studies are missing regarding the potential negative impact of biocontrol under non controlled conditions. These are easily found.
As already mentioned in the first round of review, figures are limited in quality as well as comprehensive legend. They are not informative of the major results the authors obtained. For instance figuras 1 and 2 could be improved if all the strains tested are shown and these would support the data presented in the tables. These also need better description. What does the number in Table 2 mean regarding inhibition? Material and Methods still lacks detailed information on the experimental protocol that was used and how data was collected.

Experimental design

Several relevant information are missing in the Material and Methods and these are the information of which Xoo and Bo strains were used. Specifically, the Controlled-conditions experiment lacks information regarding number of bacterial cells in pathogen-infested soil, the number of initial bacterial cells in treatments when diluted to OD 600 nm = 0.5. Finally, no details in how parameters were defined for each of the measurements such as disease incidence, biocontrol efficiency.

Validity of the findings

The biocontrol area is an emerging sustainable approach to disease control. However, very precise measurements and controlled experiments are needed to address the efficacy of such procedures and how sustainable they are over different places of the globe.

Reviewer 4 ·

Basic reporting

This study impactfully describes the role of rice as a major stapple food around the globe and yield-lowering impact of phytopathogens it faces. Chemical pesticides though offering immediate relief but harmful for the environment. On the other hand, bioinoculants as biofertilizers and biopesticides significantly ensures the environmental safety and crop yield. Authors isolated a large set of rice root associated bacteria from multiple rice cultivation sites of Pakistan and assessed their beneficial characteristics for rice growth and disease-protection. Using several in vitro and in vivo testing, they found that bacteria isolates, LE7 and LR22, were highly effective to improve rice growth and control rice major pathogens/disease. Furthermore, authors identified taxonomically most effective isolates by 16S rRNA sequence technique. Manuscript also highlights the first report of occurrence of Bacillus amyloliquefaciens in rice rhizosphere. The study is methodologically legit and impactful in the field, however minor issues like word formation errors and typo-mistakes need to be corrected before publication.

Experimental design

No comment

Validity of the findings

No comment

Additional comments

Page 2 L 27 “plant pathogen”, replace it with ‘plant pathogens’
Page 3 L 54 “broad-range”, replace it with more suitable word like “non-specific”
Page 3 L 66 “highly-demanded”, remove the hyphen between two words
Page 5 L 96 “Six-weeks”, change it to “six-week”
Page 5 L 96: kindly mention if you used any medium for the storage of rice root samples
Page 5 L 99: Remove comma from “Shahid et al.,”
Page 6 L 114: Kindly mention if flasks or tubes were used to store pure cultures in 20% glycerol at -80 ºC
Page 7 L 146: change “2 minutes” to 2 “min”
Page 7 L 148: replace “OD600 nm = 0.2” with “0.2 OD600 nm”
Page 8 L 167: remove comma from ‘Tariq et al., (2010)’
Page 10 L 212: Correct the spelling of ‘La Pierie et al. (2017)’
Page 10 L 224 “similarity”, change it to “similarities”
Page 11 L 229: “A total number of”, change it to “a total of”
Page 13 L 272: Rephrase the sentence “Based on the better biocontrol and plant growth-promoting activities, two bacterial isolates, LR22 and LE7, were selected to be identified by sequencing their 16S ribosomal RNA gene and comparing the sequences against NCBI databases.”
Page 14 L 295-296: arrange citation year-wise and remove the bracket of Zhan et al., (2023).
Page 15 L 338: Italicize the ‘Sasamorpha borealis’
Page 16 L 343: remove comma from ‘Nigris et al., (2018)’
Page 16 L 347: remove comma from ‘Hassan et al., (2019)’
Page 17 L 384: remove both commas from ‘Rawat et al., (2022),’. Check such mistake throughout the manuscript.

---

## Round 0.3 · accepted · Accept

The authors have satisfactorily addressed all the queries raised by the reviewers. I have gone through the responses and found satisfactory. The manuscript is now ready for publication in its current form.